DATA RELEASE

# The genome assembly and annotation of the Oriental rat snake *Ptyas mucosa*

Jiangang Wang[1,2,†], Shiqing Wang[2,3,†], Song Huang[4], Qing Wang[2,3], Tianming Lan[2], Ming Jiang[1], Haitao Wu[1,*] and Yuxiang Yuan[1,*]

1 Key Laboratory of Wetland Ecology and Environment & Heilongjiang Xingkai Lake Wetland Ecosystem National Observation and Research Station, Northeast Institute of Geography and Agroecology, Chinese Academy of Sciences, 130102, Changchun, China
2 State Key Laboratory of Agricultural Genomics, BGI-Shenzhen, Shenzhen, 518083, China
3 College of Life Sciences, University of Chinese Academy of Sciences, Beijing, 100049, China
4 Anhui Province Key Laboratory of the Conservation and Exploitation of Biological Resource, College of Life Sciences, Anhui Normal University, Wuhu, 241000, China

## ABSTRACT

The Oriental rat snake *Ptyas mucosa* is a common non-venomous snake of the colubrid family, spanning most of South and Southeast Asia. *P. mucosa* is widely bred for its uses in traditional medicine, scientific research, and handicrafts. Therefore, genome resources of *P. mucosa* could play an important role in the efficacy of traditional medicine and the analysis of the living environment of this species. Here, we present a highly continuous *P. mucosa* genome with a size of 1.74 Gb. Its scaffold N50 length is 9.57 Mb, and the maximal scaffold length is 78.3 Mb. Its CG content is 37.9%, and its gene integrity reaches 86.6%. Assembled using long-reads, the total length of the repeat sequences in the genome reaches 735 Mb, and its repeat content is 42.19%. Finally, 24,869 functional genes were annotated in this genome. This study may assist in understanding *P. mucosa* and supporting medicinal research.

**Submitted:** 18 May 2023

\* Corresponding authors. E-mail: wuhaitao@iga.ac.cn; yuanyuxiang@iga.ac.cn

† Contributed equally.

Preprint submitted at https://doi.org/10.20944/preprints202308.1451.v1

Included in the series: ***Snake Genomes*** (https://doi.org/10.46471/GIGABYTE_SERIES_0004)

**Subjects** Genetics and Genomics, Evolutionary Biology, Zoology

## INTRODUCTION

Known as the Oriental rat snake (Figure 1) [1], Indian rat snake, or Dhaman, *Ptyas mucosa* is a common non-venomous species of colubrid snakes. There are over 300 genera and 2,000 species in the colubrid family, making it the largest snake family [2]. While an excitable and fast-moving snake, the rat snake is harmless to humans, preying upon small reptiles, birds, and mammals. Therefore, in some areas, farmers obtain the Oriental rat snake from other locations to catch mice and protect their crops. Adult snakes usually prefer to subdue their prey by sitting on it instead of constricting it, using their weight to overpower it, a hunting mechanism for capturing prey seldom observed in other snake species [3]. When threatened, Oriental rat snakes inflate their necks, imitating the king cobra or Indian cobra to scare potential predators [4].

In southern China, the Indian rat snake is commonly eaten by humans, and its skin is used for making the membranes of a traditional musical instrument, the erhu [5]. Traditional Chinese medicine uses its gallbladder to prepare a medicinal wine for treating many diseases [6]. In the past, due to overhunting, its number was significantly reduced; however, artificial breeding succeeded in gradually recovering their number [6].

**Figure 1.** A picture of a *P. mucosa* individual, by Probophilic CC0 Wikimedia commons.

In this study, we present a highly continuous genome of *P. mucosa* with a genome size of 1.74 Gb. The genome was generated using single-tube long fragment reads (stLFR) sequencing data, combined with whole genome sequencing data for correction. Its repeat content reached 42.19%. This genome is an important basis for follow-up studies elucidating the biology *of P. mucosa*. In particular, high-quality reference genome and transcriptome data can provide effective help for subsequent targeted breeding.

## MAIN CONTENT

### Context

In this study, we present a highly-continuous genome assembly of *P. mucosa*. The maximum genome size is 1.74 Gb. The length of scaffold N50 is 9.57 Mb, and the maximal length of the scaffold is 78.3 Mb (Table 1). Furthermore, our *P. mucosa* genome has a CG content of 37.9% and, using BUSCO (v5.2.2; RRID:SCR_015008) (Figure 2), we found that its integrity reaches 86.6%. Thus, according to these genome assembly data, this is a highly contiguous genome. Here, we report the draft reference genome sequence of *P. mucosa*. This data will be a valuable resource in the study of non-poisonous snakes.

### Methods

Detailed stepwise protocols are gathered in a protocols.io collection with the minor adaptations outlined below [7] (Figure 3).



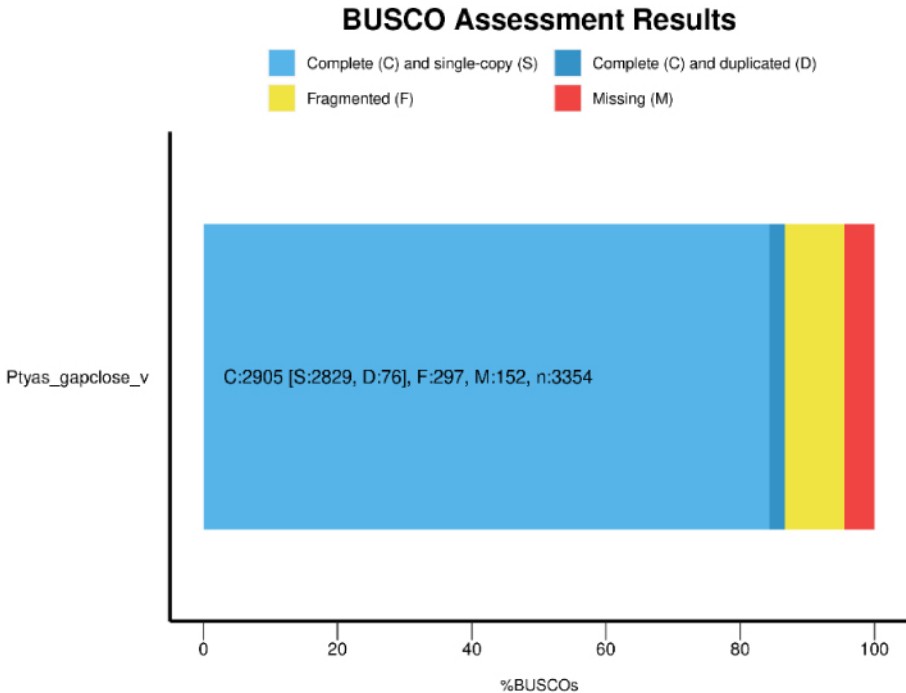

**Figure 2.** BUSCO assessment result of our *P. mucosa* genome.

**Table 1.** Summary of the features of our *P. mucosa* genome.

| | Contig | Scaffold |
|---|---|---|
| Maximal length (bp) | 317,010 | 78,354,666 |
| N90 (bp) | 4,639 | 10,835 |
| N50 (bp) | 23,622 | 9,579,637 |
| Number ≥ 100 bp | 189,926 | 87,170 |
| Number ≥ 2 kb | 110,746 | 35,256 |
| GC content (%) | 40.3 | 37.9 |
| Genome size (bp) | 1743610025 | |

## Sample collection and sequencing

In 2021, an adult *P. mucosa* (NCBI:txid31142) individual from Hezhou City in the Guangxi province of China was collected for genome assembly and RNA sequencing. The snake was identified as *P. mucosa* by morphological identification. The individual died of natural causes and its samples were transferred to dry ice, quickly frozen, and kept at −80 °C until further use. We isolated eight tissues and organs for RNA sequencing, including the heart, the small intestine, the large intestine, the lung, the liver, the stomach, the kidney, and the muscles. Furthermore, genomic DNA was extracted for whole-genome sequencing utilizing the AxyPrep genomic DNA kit (AxyPrep, USA).

The total RNA was isolated utilizing the TRlzol reagent (Invitrogen, USA) following the recommended guidelines. The RNA quality, purity, and quantity were assessed using a Qubit 3.0 fluorometer (Life Technologies, USA) and an Agilent 2100 Bioanalyzer System (Agilent, USA). The cDNA libraries were generated through the reverse transcription of RNA fragments ranging from 200 to 400 bp. In addition, the liver sample was used for stLFR sequencing and genome survey. The latter refers to methods for analyzing second

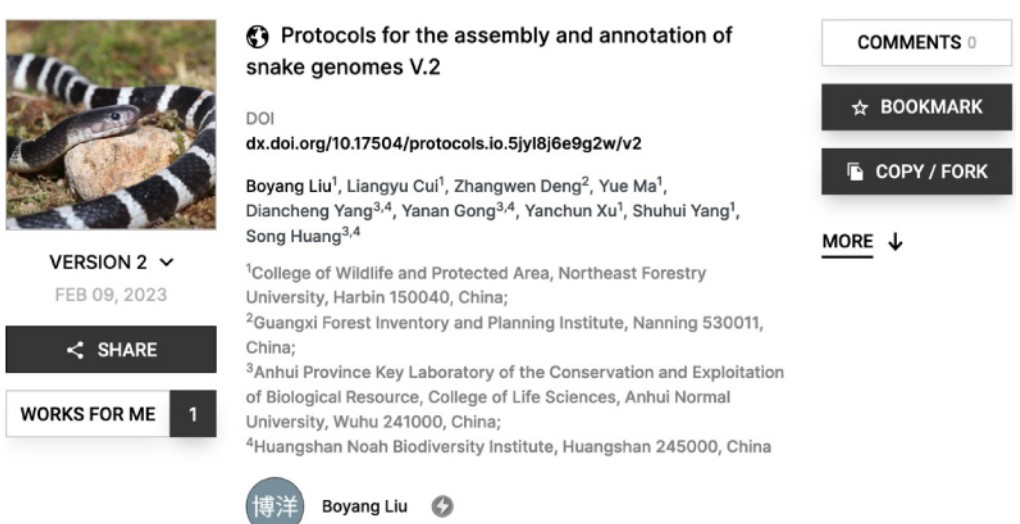

**Figure 3.** A protocols.io collection of the standard protocols for sequencing snake genomes [7]. https://www.protocols.io/widgets/doi?uri=dx.doi.org/10.17504/protocols.io.5jyl8j6e9g2w/v2

generation sequencing data through k-mer to obtain genome size, heterozygosity, repeat sequence proportion, GC-content, and other genomic information.

## Genome survey, assembly, annotation, and assessment

The stLFR sequencing data were assembled using Supernova (v2.1.1, RRID:SCR_016756) [8]. NextPolish (v1.0.5) [9] was then used to perform a second round of correction and a third round of polishing of this assembly using the Whole Genome Sequencing (WGS) data. To get a haploid representation of the genome, duplicates were purged from the genome using the purge_dups pipeline (RRID:SCR_021173) [10]. The completeness of the genome was evaluated using sets of BUSCO (v5.2.2) with genome mode and lineage data from vertebrata_odb10 [11].

In order to detect the presence of known repeat elements in the genome of the many-banded *P. mucosa*, the following approach was employed. To identify the known repetitive elements in the genome of the many-banded krait, Tandem repeats Finder [12], LTR_Finder (RRID:SCR_015247) [13], and RepeatModeler (v2.0.1, RRID:SCR_015027) [14] were employed for the purpose. RepeatMasker (v3.3.0, RRID:SCR_012954) [15] and RepeatProteinMask v3.3.0 [16] were used to search the genome sequences for known repeat elements. The BRAKER2 pipeline (RRID:SCR_018964) [17] was used for gene prediction. Then, the gene sets were aligned against several known databases, including SwissProt [18], TrEMBL [18], Kyoto encyclopedia of genes and genomes (KEGG) [19], gene ontology (GO), and the NR [20] database.

## Results

In *P. mucosa*, the total length of the repeat sequence in the genome reaches 735 Mb, and its repeat content is as high as 42.16% (Tables 2 and 3). We analysed the content of various repetitive elements, and several different genome families were identified within the *P. mucosa* genome. We found that long interspersed nuclear elements (LINEs) accounted for 35.51%, long terminal repeat (LTR) accounted for 9.15%, and DNA accounted for 4.66%



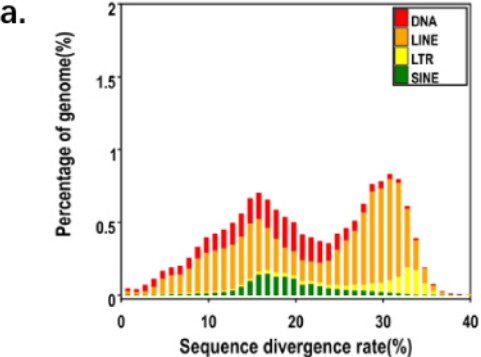
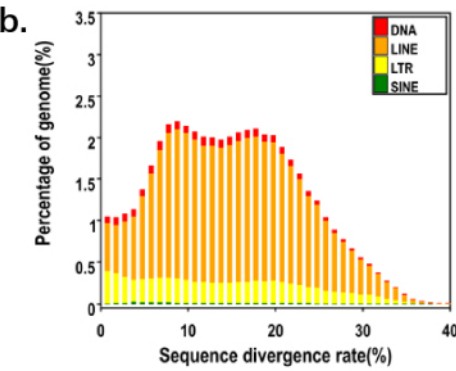

**Figure 4.** Distribution of TEs in our *P. mucosa* genome. TEs include DNA transposons (DNA) and RNA transposons (i.e., DNAs, LINEs, LTRs, and short interspersed nuclear elements (SINEs)). (a) Known sequence divergence rate distribution. (b) *De novo* sequence divergence rate distribution.

**Table 2.** Statistics for the repetitive sequences identified in our *P. mucosa* genome.

| Type | Length (bp) | % in genome |
|---|---|---|
| DNA | 41,761,899 | 2.395689 |
| LINE | 581,624,764 | 33.365146 |
| SINE | 8,061,060 | 0.462426 |
| LTR | 149,994,747 | 8.604511 |
| Other | 0 | 0 |
| Satellite | 2,433,786 | 0.139615 |
| Simple_repeat | 10,136,004 | 0.581456 |
| Unknown | 5,653,213 | 0.324299 |
| Total | 735,004,828 | 42.163857 |

**Table 3.** Summary of TEs in our *P. mucosa* genome.

| Type | Repbase TEs | | TE proteins | | *De novo* | | Combined TEs | |
|---|---|---|---|---|---|---|---|---|
| | Length (bp) | % in genome | Length (bp) | % in genome | Length (bp) | % in genome | Length (bp) | % in genome |
| DNA | 39,281,826 | 2.35 | 6,433,176 | 0.38 | 37,917,702 | 2.26 | 71,410,039 | 4.27 |
| LINE | 186,209,051 | 11.14 | 150,758,176 | 9.02 | 449,338,074 | 26.89 | 511,842,308 | 30.63 |
| SINE | 20,280,301 | 1.21 | 0 | 0 | 2,779,035 | 0.16 | 22,466,386 | 1.34 |
| LTR | 34,138,399 | 2.04 | 53,662,430 | 3.21 | 224,765,038 | 13.45 | 234,525,215 | 14.03 |
| Other | 25,447 | 0.002 | 0 | 0 | 0 | 0 | 25,447 | 0.002 |
| Unknown | 0 | 0 | 0 | 0 | 7,924,824 | 0.47 | 7,924,824 | 0.47 |
| Total | 266,507,708 | 15.95 | 210,726,751 | 12.61 | 667,082,033 | 39.92 | 705,048,693 | 42.19 |

(Figure 4). Hence, LINEs were the most frequent repeats. Despite snake species sharing similar genome sizes, research findings demonstrated considerable variations in transposable element (TE) content, with limited diversity in the types of TEs. In particular, species with a longer evolutionary history tend to exhibit greater diversity in TE content, as indicated by research findings.

A total of 24,869 functional genes were annotated using KEGG. This showed the highest number of annotated genes in pathways related to Human Diseases, Organismal Systems, and Metabolism. The highest number of Signal Transduction genes were found in Environmental Information Processing. Moreover, our GO gene enrichment for *P. mucosa* revealed that, among 25 biological process pathways, 247 genes related to immune system processes, and two genes related to detoxification (Figure 5).



**Figure 5.** Gene annotation information of *P. mucosa*. (a) KEGG enrichment of *P. mucosa*. (b) GO enrichment of *P. mucosa*.

## REUSE POTENTIAL

*P. mucosa* is a species of snake belonging to the species-rich Colubrid family. Therefore, assembling the genome of *P. mucosa* helps understand the development process and the origin of Colubrids. Alongside this, as an economically important species, understanding the genome of *P. mucosa* can potentially guide the breeding of the rat snake. While working on this genome, we discovered that another *P. mucosa* genome has been posted in GenBank (GCA_012654045.1). As our genome was assembled using stLFR data, it is also potentially useful to have genomes of different individuals and using different sequencing technologies available to enable comparisons of different populations and sequencing technologies.

## DATA AVAILABILITY

The data supporting the findings of this study was deposited into the CNGB Sequence Archive of the China National GeneBank DataBase with the accession number CNP0004141. Raw reads are available in the SRA via bioproject PRJNA955401, and additional data is in the GigaDB repository, such as protein data, BUSCO comparison data, annotation data, etc. [21].

## EDITOR'S NOTE

This paper is part of a series of Data Release papers presenting the genomes of different snake species [22].

## ABBREVIATIONS

BUSCO, Benchmarking Universal Single-Copy Orthologs; GO, Gene Ontology; KEGG, Kyoto encyclopedia of genes and genomes; LINEs, long interspersed nuclear elements; LTR, long terminal repeat; SINE, short interspersed nuclear element; stLFR, single-tube long fragment reads; TE, transposable element.

## DECLARATIONS

### Ethics approval and consent to participate

The authors declare that ethical approval was not required for this type of research.

### Consent for publication

Not applicable.

### Competing Interests

The authors declare no conflict of financial interests.

### Author contribution

TL designed and initiated the project. JW performed the DNA extraction, the library construction, and the data analysis, and wrote the manuscript. All authors read and approved the final manuscript.

## Funding

Our project was financially supported by the Guangdong Provincial Key Laboratory of Genome Read and Write (grant no. 2017B030301011). This work was also supported by China National GeneBank (CNGB).

## Acknowledgements

The samples were collected by the Northeast Institute of Geography and Agroecology, Chinese Academy of Sciences.

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
