## [Reviewer Report]

Comments on revised manuscriptThis study presents a haploid genome assembly of Ptyas mucosa. The authors' chosen approach for genome assembly appears valid, yielding a reasonable outcome, albeit not at the chromosomal level. Furthermore, this study has conducted a comprehensive analysis of genomic features, yielding compelling results. In my opinion, the study meets the requirements for publication.

---

## [Editor Report]

Editor’s AssessmentThe Oriental ratsnake Ptyas mucosa is a common non-venomous snake with a wide geographic range spanning much of South and Southeast Asia.. To help better understand the evolution of P. mucosa, and also better understand its use in traditional medicine a 1.74Gb in size reference genome was sequenced and described in this work. This data can be combined with already published and upcoming snake genome data to construct the evolutionary history of snakes and other reptiles. After submission, a second P. mucosa genome has been deposited in NCBI, but it is useful to have data from multiple individuals, and using different sequencing technologies (in this case stLFR).

---

## [Reviewer Report]

Upload additional filesDRR-202305-05/form/reviews.docxReviewer name and names of any other individual's who aided in reviewer Zexian ZhuDo you understand and agree to our policy of having open and named reviews, and having your review included with the published papers. (If no, please inform the editor that you cannot review this manuscript.)YesIs the language of sufficient quality?YesPlease add additional comments on language quality to clarify if needed
Are all data available and do they match the descriptions in the paper? YesAdditional CommentsAre the data and metadata consistent with relevant minimum information or reporting standards? See GigaDB checklists for examples <a href="http://gigadb.org/site/guide" target="_blank">http://gigadb.org/site/guide</a>YesAdditional CommentsIs the data acquisition clear, complete and methodologically sound?YesAdditional CommentsIs there sufficient detail in the methods and data-processing steps to allow reproduction?NoAdditional Commentssome parts of the methods are not clear enoughIs there sufficient data validation and statistical analyses of data quality? YesAdditional CommentsIs the validation suitable for this type of data?YesAdditional CommentsIs there sufficient information for others to reuse this dataset or integrate it with other data?YesAdditional CommentsAny Additional Overall Comments to the AuthorSee the attachment.RecommendationMajor Revision

---

## [Reviewer Report]

Upload additional filesDRR-202305-05/form/GIGABYTE-DRR-TEMP-1684395677-Wang et al.pdfReviewer name and names of any other individual's who aided in reviewer Merly EscalonaDo you understand and agree to our policy of having open and named reviews, and having your review included with the published papers. (If no, please inform the editor that you cannot review this manuscript.)YesIs the language of sufficient quality?YesPlease add additional comments on language quality to clarify if needed
Are all data available and do they match the descriptions in the paper? NoAdditional CommentsAre the data and metadata consistent with relevant minimum information or reporting standards? See GigaDB checklists for examples <a href="http://gigadb.org/site/guide" target="_blank">http://gigadb.org/site/guide</a>YesAdditional CommentsIs the data acquisition clear, complete and methodologically sound?NoAdditional CommentsSee attached documentIs there sufficient detail in the methods and data-processing steps to allow reproduction?YesAdditional CommentsAlthough some values are missing and can be inferred.Is there sufficient data validation and statistical analyses of data quality? YesAdditional CommentsThere's enough validation for the genome assembly. I can't be certain of the annotation since is not my area of expertise.Is the validation suitable for this type of data?YesAdditional CommentsIs there sufficient information for others to reuse this dataset or integrate it with other data?YesAdditional CommentsAny Additional Overall Comments to the AuthorRecommendationMajor Revision

---

## [Reviewer Report]

Reviewer name and names of any other individual's who aided in reviewer Takushi KishidaDo you understand and agree to our policy of having open and named reviews, and having your review included with the published papers. (If no, please inform the editor that you cannot review this manuscript.)YesIs the language of sufficient quality?NoPlease add additional comments on language quality to clarify if needed
Are all data available and do they match the descriptions in the paper? YesAdditional CommentsAre the data and metadata consistent with relevant minimum information or reporting standards? See GigaDB checklists for examples <a href="http://gigadb.org/site/guide" target="_blank">http://gigadb.org/site/guide</a>YesAdditional CommentsIs the data acquisition clear, complete and methodologically sound?YesAdditional CommentsIs there sufficient detail in the methods and data-processing steps to allow reproduction?NoAdditional CommentsIs there sufficient data validation and statistical analyses of data quality? YesAdditional CommentsIs the validation suitable for this type of data?YesAdditional CommentsIs there sufficient information for others to reuse this dataset or integrate it with other data?YesAdditional CommentsAny Additional Overall Comments to the AuthorThis study provides a haploid genome assembly of an oriental ratsnake Ptyas mucosa. I think the genome assembling approach taken by the authors is valid, and the obtained assembly is reasonable, although it is not in the chromosomal-level. However, a genome assembly of this species with better statistics (e.g., higher N50 value) has already been provided in GenBank with accession no. GCA_012654045.1. The authors should mention about this assembly, and should state why they determined to provide another genome assembly of the same species.  Other comments are as follows: Total sequencing amount, sequencing length, sequencing platform, and sequencing design (paired or single) should be provided for each DNA/RNA sequencing. A sentence in the “Genome survey, assembly, annotation and assessment” section: “Next, Repeat Finder (TRF) …” is incomplete. References are required for algorithms and applications used in this study such as “stLFR sequencing”, “arrow algorithm”, and “NextPolish”. Table 1: Because contig does not contain unknown bases (N), contig genome size should be smaller than that of scaffolds. I think Fig. 3 is not required and should be deleted.
RecommendationMajor Revision